# Seafloor observations indicate spatial separation of coseismic and postseismic slips in the 2011 Tohoku earthquake

Takeshi Iinuma[1], Ryota Hino[2], Naoki Uchida[2], Wataru Nakamura[2,†], Motoyuki Kido[3], Yukihito Osada[2,†] & Satoshi Miura[2]

Large interplate earthquakes are often followed by postseismic slip that is considered to occur in areas surrounding the coseismic ruptures. Such spatial separation is expected from the difference in frictional and material properties in and around the faults. However, even though the 2011 Tohoku Earthquake ruptured a vast area on the plate interface, the estimation of high-resolution slip is usually difficult because of the lack of seafloor geodetic data. Here using the seafloor and terrestrial geodetic data, we investigated the postseismic slip to examine whether it was spatially separated with the coseismic slip by applying a comprehensive finite-element method model to subtract the viscoelastic components from the observed postseismic displacements. The high-resolution co- and postseismic slip distributions clarified the spatial separation, which also agreed with the activities of interplate and repeating earthquakes. These findings suggest that the conventional frictional property model is valid for the source region of gigantic earthquakes.

[1] Research and Development Center for Earthquake and Tsunami, Japan Agency for Marine-Earth Science and Technology, Yokohama 236-0001, Japan. [2] Research Center for Prediction of Earthquakes and Volcanic Eruptions, Graduate School of Science, Tohoku University, Sendai 980-8578, Japan. [3] Disaster Science Division, International Research Institute of Disaster Science, Tohoku University, Sendai 980-0845, Japan. † Present addresses: Japan Meteorological Agency, Tokyo 100-8122, Japan (W.N.); GNSS Technologies Inc., Tokyo 160-0022, Japan (Y.O.). Correspondence and requests for materials should be addressed to T.I. (email: iinuma@jamstec.go.jp).

Both seismic (unstable) and aseismic (stable) slip can occur on the plate interfaces at convergent[1,2] and transform plate boundaries[3–5], and it has been broadly accepted that these two different types of slip show complementary spatial distributions[2–7]. The differences in slip behaviours have been explained as the consequence of variations in the frictional parameters of the empirical rate- and state-dependent friction law[8,9], which are expected to be strongly variable along a fault. Under ordinary circumstances, those parts of a fault hosting a seismic rupture are unable to exhibit much stable sliding in response to the external stress associated with the secular plate motion. Thus, quasi-static postseismic slip occurs mainly in those areas shallower and/or deeper than the coseismic rupture zone. Sometimes postseismic slip propagates along the fault strike direction and promoted large earthquakes at neighbouring seismic patches[10–13]. This conceptual model has been supported by a number of studies that have revealed postseismic slip distributions based on geophysical observations.

Previous studies on the rupture process[14–21] and coseismic slip distribution[22–25] of the M9.0 2011 Tohoku Earthquake occurred on 11 March 2011 have consistently revealed that substantial amount of coseismic slip of ∼10 m occurred on the deep (depth: 40–50 km) potion of the plate interface as well as the extremely large (>30 m) coseismic slip on the shallow (depth: <30 km) fault, although the models show significant diversities in detailed slip pattern reflecting the differences in the data, analysis procedure and assumed structure[14–25]. The deeper portion of the mainshock rupture included the source region of the Miyagi-Oki earthquakes, a sequence of repeating ∼M7.5 interplate earthquakes with a recurrence interval of ∼40 years. The most recent event in the sequence happened on 12 June 1978 (Mw7.5) (refs 25,26).

The postseismic slip associated with the 2011 Tohoku Earthquake has been estimated using continuous global positioning system (GPS) observations and the activities of smaller repeating earthquakes[27–32]. Some studies reported that co- and postseismic slip overlap[27–32] and a possibility of breakdown of the complementary distributions of the slips was suggested and a doubt regarding applicability of the empirical rate- and state-dependent friction law to the natural earthquake faults was raised[28]. Although the spatial resolutions of the estimations of co-, post- and/or interseismic slip distributions are limited without sufficient near-field observations, these analyses did not include seafloor geodetic data. Such data can provide a strong constraint for the estimation of a component of the postseismic deformation due to the viscoelastic relaxation process in the asthenosphere, which occurs simultaneously with the postseismic slip on the plate interface. Recently, landward motion of the seafloor above the main rupture area of the Tohoku Earthquake has been reported, which constitutes strong evidence for the larger contribution of the viscoelastic relaxation[33,34].

In the present study, we investigated the spatial distribution of the postseismic slip on the plate interface. This was based on both terrestrial GPS data and seafloor geodetic observations composed of GPS/Acoustic (GPSA) survey results and ocean-bottom pressure (OBP) gauges records. Among these observations, the continuous time series of vertical seafloor displacement provided by the OBP recordings, newly reported in this study, was expected to improve considerably the spatial resolution of the postseismic slip on the plate interface. By taking into account the viscoelastic relaxation, the estimated postseismic slip distribution is consistent with the activities of interplate and repeating earthquakes. The broad area of the coseismic slip during the M9 mainshock had exhibited different behaviours during the previous smaller earthquakes or episodic slow slip events, but does not overlap with the zone of the aseismic slip after the 2011 mainshock. The result suggests that the conventional frictional property model based on the rate- and state-dependent friction law is valid also for the source region of the gigantic earthquake.

## Results

**Postseismic slip based on geodetic data.** Figure 1 shows the cumulative postseismic slip distribution, computed for the analysis period of ∼8 months, together with its estimation error. Comparisons between the observed displacements and those calculated (predicted) based on the estimated postseismic slip model, and their residuals, are shown in Figs 2 and 3, respectively. The results reveal that the area in which large postseismic slip occurred on the plate interface can be generally divided into three subareas: (1) the intermediate to deep (depth: 25–55 km) portion in the latitude range of 38.5–40.2°N (corresponding to off Iwate Prefecture); (2) the intermediate (depth: 30–40 km) portion in the latitude range of 37.5–38.1°N (corresponding to off southern Miyagi and northern Fukushima prefectures); and (3) the shallow (depth: <20 km) portion in the latitude range of 35.0–36.5°N (corresponding to off southern Fukushima and Ibaraki prefectures). These three subareas of substantial postseismic slip are located beyond the distribution of coseismic slip during the 2011 Tohoku Earthquake, which was estimated based on geodetic data obtained by almost the same seafloor and terrestrial observation networks[25] (Fig. 1).

The residual horizontal displacements show evident along-strike motion. These may have arose from the strike-slip components of the postseismic slip, the effect of the subducting Philippine Sea slab to the viscoelastic relaxation, the error due to the transformation from spheroidal to Cartesian coordinates, and/or uncertainty of the plate motion model to transform the displacements into Okhotsk-plate-fixed reference frame.

**Postseismic slip based on seismic data.** The spatial extent of the postseismic slip can be constrained by analysis of the small repeating earthquakes that occur on the plate interface[29]. Figure 4 shows the estimated postseismic slip distribution, based on the activities of small repeating earthquakes (see ref. 29 for the analysis procedure), for the same period as the geodetic inversion analysis in the present study (Fig. 1). The estimated slip distribution clarifies that no significant postseismic slip occurred in the area of the mainshock rupture.

Analysis of the small repeating earthquakes enable us to estimate the rate of aseismic slip around small isolated seismic patches (asperities) on the plate interface causing the small repeating earthquakes[35]. However, we cannot evaluate the slip rate in the region near the Japan Trench, to the north and south of the large coseismic rupture area, because no asperities generating repeating earthquakes have been identified in the areas. Instead, we focused on the activity of the interplate earthquakes other than the repeating earthquakes, assuming that their activity would be increased in association with the acceleration of aseismic slip, as is usually assumed in the studies of repeating earthquakes. Although it is difficult to estimate the slip rate quantitatively based on the non-repeating earthquakes, their activation can be interpreted as the acceleration of aseismic slip in the area. Figure 5 shows the ratio of seismicity rate during the period in which postseismic slip was estimated to that during the preseismic period from the beginning of 2008 to just before the mainshock based on the catalogue of the thrust faulting earthquakes[36]. It clearly shows prominent activation of interplate earthquakes, in the areas surrounding the mainshock rupture, indicating the acceleration of the aseismic slip, that is, the occurrence of the postseismic slip.

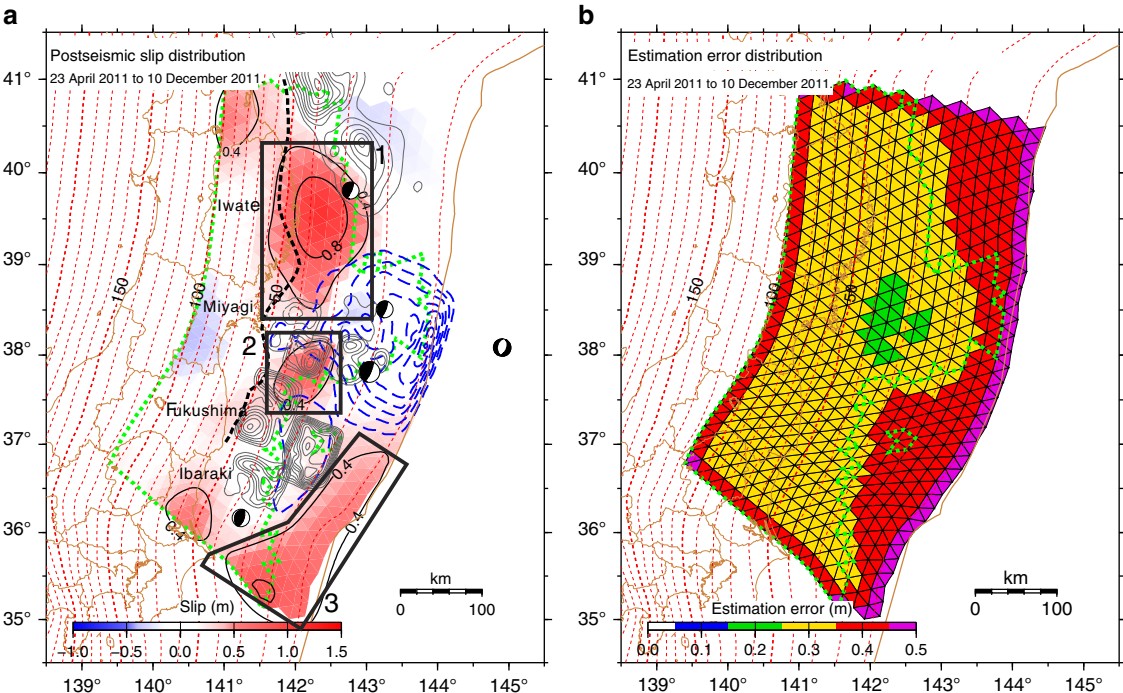

**Figure 1 | Results of the geodetic inversion analysis.** (**a**) Cumulative distribution of the estimated postseismic slip of the 2011 Tohoku Earthquake. Each triangular fault is coloured according to its dip-slip component of postseismic slip. The colour scale for the postseismic slip is displayed at the bottom of the panel. The black contours of the magnitude of slip are shown at 0.4 m intervals. The black boxes 1–3 indicate the subareas of substantial postseismic slip. The blue dashed contours represent the coseismic slip distribution[25]. The black dashed line denotes the down-dip limit of the interplate earthquakes[56]. The broken red lines show the depth of the subducting plate interface[50]. A green dotted line surrounds the area in which the average residuals between the given and estimated slips among the computational tests shown in Supplementary Fig. 3 are < 0.3 m (Supplementary Fig. 4). Grey contours show the rupture area of past large interplate earthquakes[43,57–59]. The locations of the centroids of the mainshock, largest foreshock (9 March 2011, M7.3), and three large aftershocks that occurred on 11 March 2011 (M7.4 off Iwate Prefecture, M7.5 far-off Miyagi Prefecture and M7.6 off Ibaraki Prefecture) are indicated, together with their focal mechanisms as determined by the Japan Meteorological Agency. Names of prefectures included in the body text are also displayed. (**b**) Distribution of the estimation error of the postseismic slip.

**Sensitivity of the geodetic observations to the slip near the trench**. The repeating earthquake analysis showed the postseismic slip of up to 50 cm on the trench-ward side of the subarea 1 (Fig. 4), where the resolution of the geodetic inversion was poor. We calculated the displacements due to the slip in the area at the geodetic stations (Supplementary Fig. 2a,b). The amount of the given slip was 40 cm, as large as the amount estimated by the repeating earthquakes analysis. Calculated displacements were significantly smaller (less than a half) than the residual of the inversion analysis (Fig. 3), indicating that it is not possible to identify postseismic slip in this region using the existing geodetic data.

The postseismic slip amounting to ~ 50 cm was estimated from the analysis of the repeating earthquakes on the landward side of the subarea 3 identified by the geodetic inversion (Figs 1a and 4). It would be difficult to constrain the location of the postseismic slip in the off Fukushima and Ibaraki prefectures based on the existing geodetic data, because there is only one offshore geodetic site. Displacement pattern expected from the slip on the landward area (Supplementary Fig. 2c,d) and that from the slip near the trench (Supplementary Fig. 2e,f) are similar to each other. On the other hand, we have to mind that the epicentre of the far-off earthquakes determined by the onshore seismic network data are difficult to be constrained in the dip direction. The uncertainties of the repeating earthquake epicentres result in the uncertainty of the slip location. Therefore, we regard the discrepancy between the slip distributions from the geodetic and repeating earthquake analyses is insignificant although substantial amount of the postseismic slip is required along the southern side of the large coseismic slip zone.

**Comparison with the previous models**. Co- and postseismic slip distributions based on investigations that used only the terrestrial geodetic data[27–29,31] differ significantly from our results. The most prominent difference between our results and those of previous studies is the presence or absence of overlapping of the co- and postseismic slip in the source region of the interplate ~ M7.5 Miyagi-Oki earthquakes between subareas 1 and 2. Our result shows clear spatial separation between the co- and postseismic slip. We note that the difference in the postseismic slip pattern is mainly due to the estimated amount of viscoelastic displacement. As the onshore displacements due to postseismic slip and viscoelastic relaxation occurred in almost the same direction, it is difficult to distinguish the contribution of the postseismic slip from that of the viscoelastic relaxation based only on terrestrial observations. However, the landward motion, opposite to that of the postseismic slip, which was observed on the seafloor and used in the present analysis, provides strong constraint with which to separate the two factors.

The introduction of the viscoelastic effect is essential for explaining the postseismic deformation data[33,34]; however, it is difficult to find an appropriate viscoelastic structure in the crust and upper mantle with which to model the deformation quantitatively. Although a simple layered-structure model could explain both the on- and offshore horizontal displacement fields[37], such a model is clearly inconsistent with the actual tectonics in the subduction zone and it cannot explain well the observed magnitude of seafloor subsidence. The heterogeneous

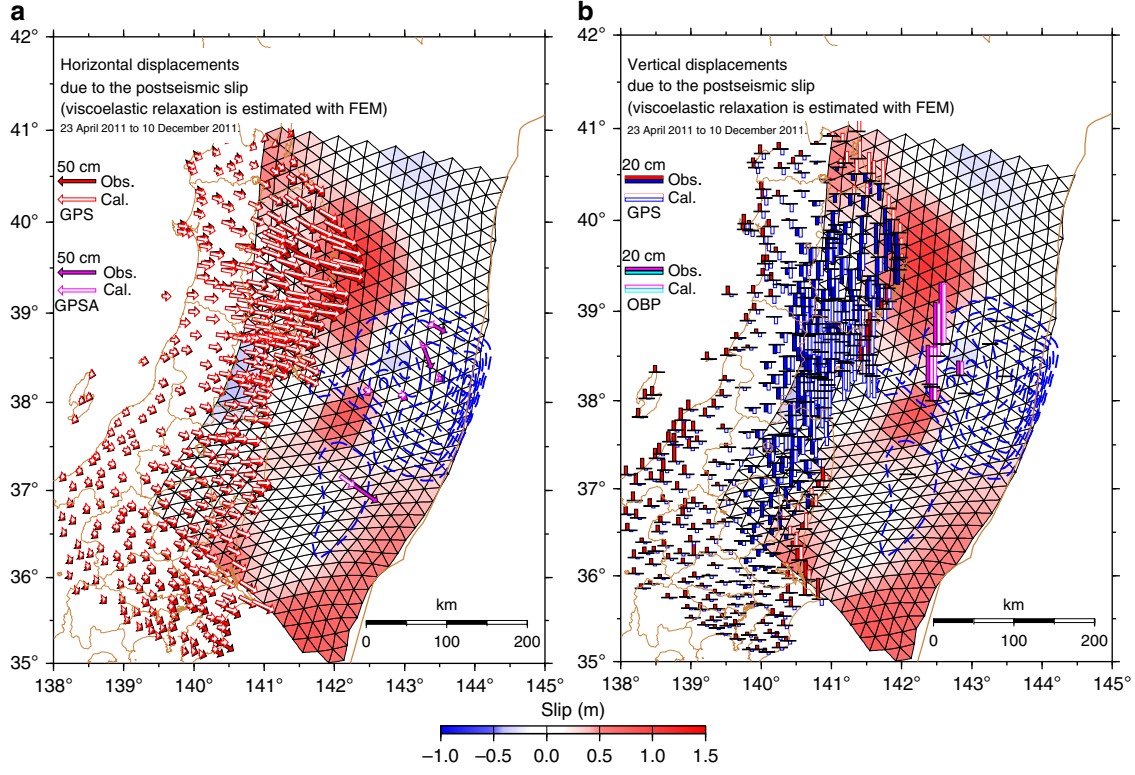

**Figure 2 | Comparison between the observed and calculated displacements.** (**a**) White and solid arrows show calculated and observed (after subtraction with respect to the aftershocks and viscoelastic relaxation) horizontal displacements. Red and magenta arrows correspond to displacements at the terrestrial GPS and seafloor GPSA sites, respectively. Cumulative distribution of the estimated postseismic slip is also shown. Each triangular fault is coloured according to its dip-slip component of postseismic slip. The colour scale for the postseismic slip is displayed at the bottom of the figure. The blue dashed contours represent the coseismic slip distribution[25]. (**b**) White and solid bars show calculated and observed (after subtraction with respect to the aftershocks and viscoelastic relaxation) vertical displacements. Red and magenta bars correspond to uplift at the terrestrial GPS and seafloor OBP sites, while blue and cyan bars correspond to subsidence at the terrestrial GPS and seafloor OBP sites, respectively.

structure including the subducting Pacific slab and cold mantle wedge[33], assumed in the present study, explains both the horizontal and the vertical deformation better than the simple layered model and it would be much more appropriate for evaluating the contribution of the viscoelastic deformation to the observed postseismic deformation.

The rheological structure and parameters estimated by ref. 33 may not be uniquely determined, but are well constrained. The viscosities controlling both the spatial distribution and temporal variation of the deformations were constrained by the time constants of decay of the observed displacement time series. The elastic thicknesses of the oceanic slab and the continental lithosphere are sensitive to the horizontal displacement pattern. Thicker oceanic slab reduces the, landward motion on the seafloor, whereas increases the seaward displacement on land. The increase of the thickness of continental lithosphere enlarges the seaward motion on land. Too large viscoelastic displacement results in the over corrections of the observed postseismic displacement field and the large normal fault type postseismic slip on the fault is required but the substantial normal faulting postseismic slip is physically unrealistic. Such trial-and-error to constrain the viscoelastic deformation model based on terrestrial and seafloor geodetic observation data were thoroughly performed by refs 33,38; therefore, we concluded that their model is the most reliable model for the present.

The simple model predicts smaller viscoelastic displacements near the Pacific coast than the novel realistic model. As an underestimation of the viscoelastic contribution results in over-estimation of the magnitude of postseismic slip, the previously estimated postseismic slip distribution could be biased, even if the viscoelastic effect were considered, as might be those based on the pure elastic assumption[33]. For example, the amount of postseismic slip beneath the northern coastal region (subarea 1) was estimated at ~1.2 m/yr, based on the analysis of the small repeating earthquakes (Fig. 4), which is much smaller than previous estimates (>5 m/yr) (refs 27,28,31). Although the slip estimation based on repeating earthquakes could be biased due to the uncertainty in the empirical scaling relationship used for the slip estimation, its consistency with that derived from the geodetic data after the careful treatment of the viscoelastic relaxation, indicates the present result is a more realistic presentation of the postseismic deformation following the 2011 Tohoku Earthquake. The argument against the spatial partitioning of seismic and aseismic behaviour on faults, based on previous geodetic data analyses, is that they could be drawn from a less reliable slip pattern and thus, we argue for the conventional concept based on an improved postseismic slip distribution.

In Supplementary Fig. 1, the estimated postseismic distribution is compared with coseismic slip distributions of various studies[14,15,17–19,23]. Spatial separation between co- and postseismic slips are well recognized in subarea 1, as in the comparisons with the slip model based on the on- and offshore geodetic data[25] (Figs 1a and 4). Although some coseismic slip distributions seem to overlap with the postseismic slip distribution at subarea 2, but we regard that the lower spatial resolutions of the slip models based on terrestrial geodetic observation and/or far-field seismic waveforms account for the apparent overlaps.

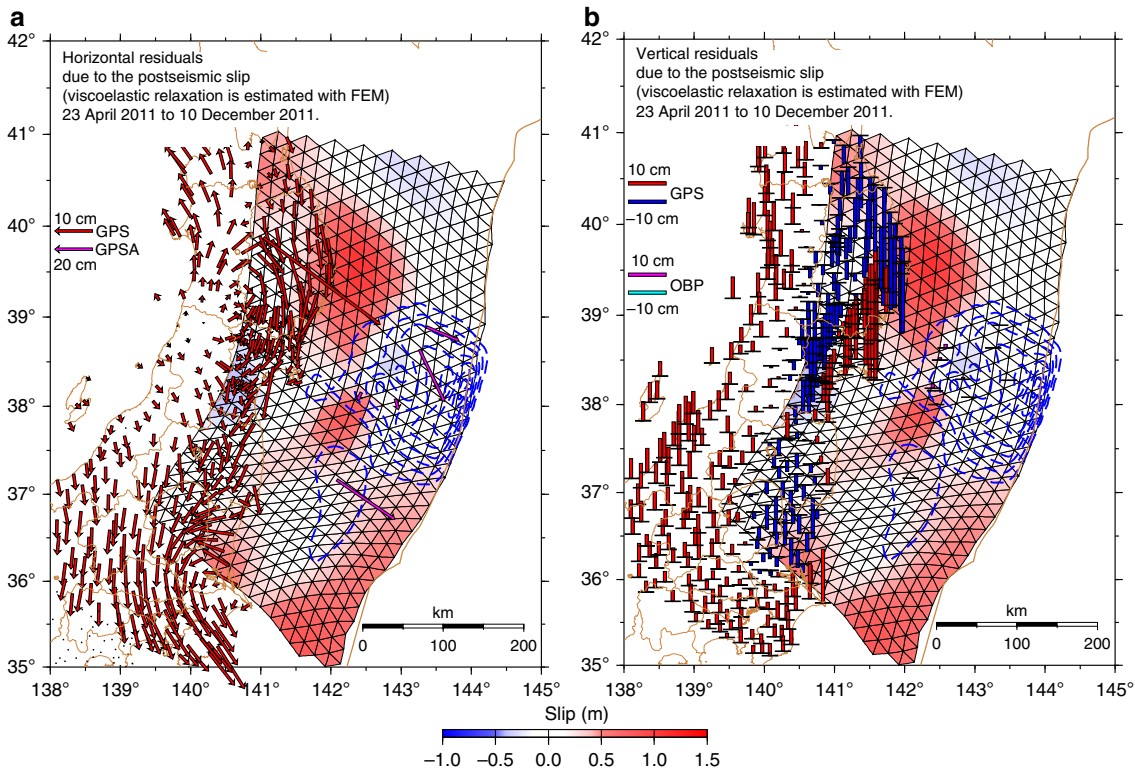

**Figure 3 | Residuals of the displacements based on the inversion analysis. (a)** Red and magenta arrows show horizontal residuals of the calculated and observed (after subtraction with respect to the aftershocks and viscoelastic relaxation) displacements (observation—calculation) at the terrestrial GPS and seafloor GPSA sites, respectively. Cumulative distribution of the estimated postseismic slip is also shown. Each triangular fault is coloured according to its dip-slip component of postseismic slip. The colour scale for the postseismic slip is displayed at the bottom of the figure. The blue dashed contours represent the coseismic slip distribution[25]. Note that the scales for the terrestrial and seafloor sites are different. **(b)** Red/blue and magenta/cyan bars show vertical residuals of the calculated and observed (after subtraction with respect to the aftershocks and viscoelastic relaxation) displacements (observation—calculation) at the terrestrial GPS and seafloor OBP sites, respectively. Red and magenta bars correspond to uplift, while blue and cyan bars correspond to subsidence.

**Spatial separation between the co- and postseismic slip.** As an important conclusion of the geodetic and seismological data analyses performed in this study, we claim that no significant postseismic slip occurred from 23 April 2011 to 10 December 2011 in the major coseismic slip zone of the mainshock area. This means that the conventional asperity model of slip behaviour[8,9] is also valid for the rupture area of the 2011 Tohoku Earthquake, the subduction zone to the northeast of Japan, and it shows that the anomalously large earthquake was not exceptional in terms of slip behaviour.

The postseismic slip is estimated in the coseismic rupture area off Fukushima and Ibaraki prefectures from the repeating earthquake analysis (Fig. 4). One may dispute that this would be a disproof of the spatial separation of co- and postseismic slip, but we regard this overlapping is apparent one caused by the lower spatial resolutions of the postseismic slip estimation as we explained early. We have to note the uncertainty of the coseismic slip in this area estimated by the geodetic data. No substantial coseismic slip was estimated if the GPSA data at the site located in the area is excluded from the inversion (see Supplementary Fig. S10 of ref. 25), meaning the coseismic slip distribution in the region is strongly dependent on the single GPSA station data. Early postseismic deformation at the site could be responsible for the mislocation of the coseismic slip because the coseismic deformation data from the campaign style GPSA surveys inevitably contain early postseismic deformation.

Although we have to refrain from making a detailed interpretation about the slip distributions in areas with low spatial resolution, it must be noted that the co- and postseismic displacements on the

seafloor off Miyagi Prefecture were well constrained by the continuous OBP records. Supplementary Fig. 4b demonstrates how the OBP records improve the spatial resolution on the slip distribution in the off Miyagi Prefecture. In the area, the coseismic slip distribution is constrained well even if the GPSA data, possibly contaminated by early postseismic deformation, are excluded[25]. Therefore, the spatial partitioning of the slip off Miyagi Prefecture is strongly supported by the present observations and analysis. Seafloor observation networks of GPSA and OBP gauges[39,40], which have been installed since the 2011 M9.0 Tohoku Earthquake and are still being developed, will contribute to future studies on the postseismic slip off the Pacific coast of Northeast Japan.

## Discussion

Spatial partitioning between the co- and postseismic slip distributions on the plate interface of the subduction zone to the northeast of Japan suggests high seismic probability at the rupture area of the 1968 Tokachi-oki Earthquake, located northeast of subarea 1. No substantial postseismic slip was identified in the rupture area of the 1968 earthquake, but the area is surrounded by extensive postseismic slip, as indicated by the geodetic data and interplate seismicity (Figs 1a, 4 and 5). These results suggest that the rupture area of the 1968 Tokachi-oki Earthquake is still coupled strongly, especially in the northern portion (40.5–41.2°N, 141.9–142.6°E) because of its stick-slip frictional property, but that the loading rate would be increased by the surrounding postseismic slip. Based on the obtained slip distribution, we argue that the occurrence time of an

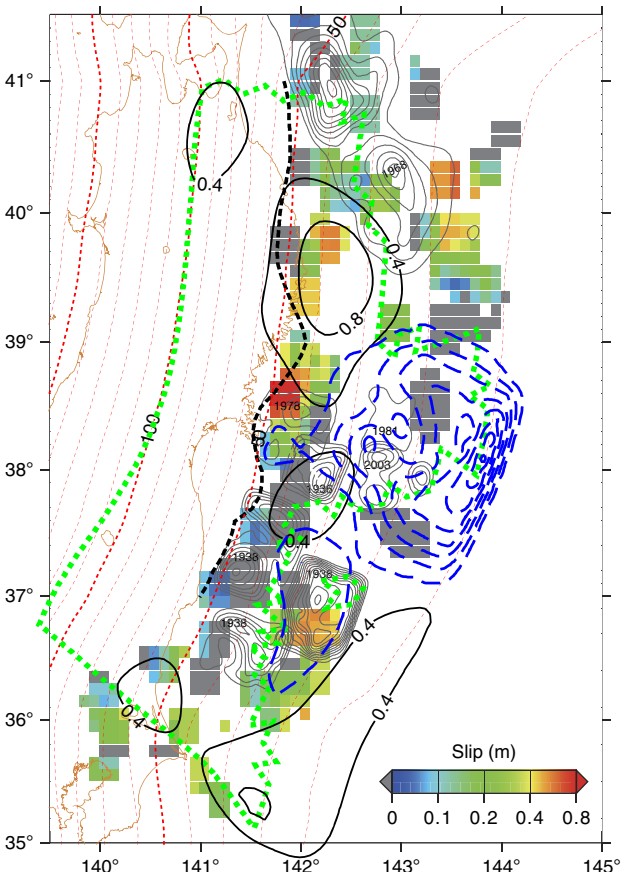
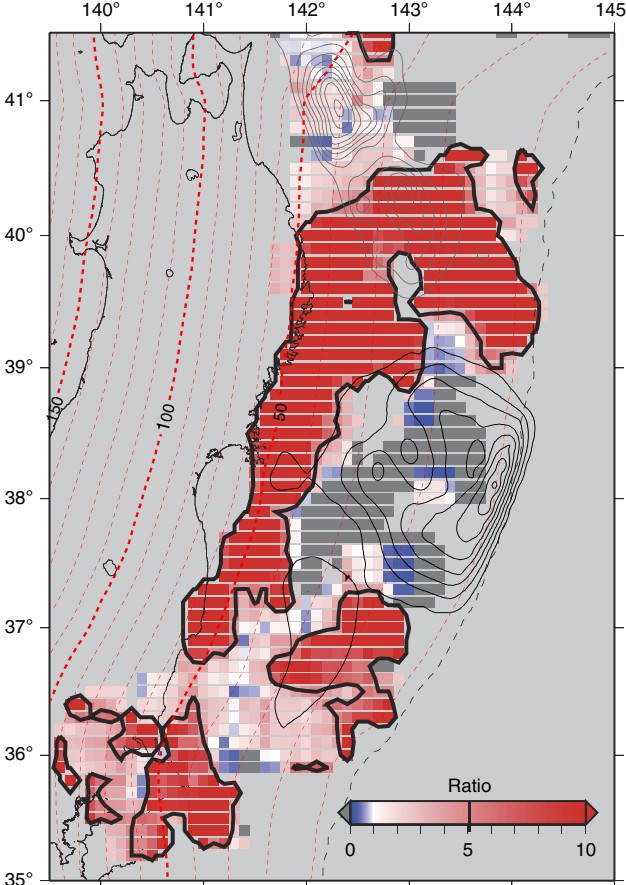

**Figure 4 | Postseismic slip from small repeating earthquakes.**
Postseismic slip distributions estimated from small repeating earthquakes
and terrestrial and seafloor geodetic data. The colour map shows the
average cumulative slip estimated from three or more repeating earthquake
groups in 0.3° × 0.3° windows, where the grey zones indicate no repeater
activity during the period of the window. Black contours represent
postseismic slip distribution based on geodetic data (identical to the black
contours in Fig. 2a). The blue dashed contours represent the coseismic slip
distribution[25] The black dashed line denotes the down-dip limit of the
interplate earthquakes[56]. The broken red lines show the depth of the
subducting plate interface[50]. A green dotted line surrounds the area in
which the average residuals between the given and estimated slips among
the computational tests shown in Supplementary Fig. 3 are <0.3 m
(Supplementary Fig. 4). Grey contours show the rupture area of past large
interplate earthquakes[43,57–59].

**Figure 5 | Acceleration of aseismic slip based on the thrust faulting
earthquake activities.** Ratio of occurrence rates (number of earthquake in
unit time) of interplate earthquakes for the period after (23 April 2011 to 10
December 2011) and before (1 January 2008 to 11 March 2011) the 2011
Tohoku Earthquake. Thick black solid contours delineate the areas, in which
the rate is five times (or more) than during the preseismic period. Thin
black contours represent the coseismic slip distribution[25]. The broken red
lines show the depth of the subducting plate interface[50]. Grey contours
show the rupture area of 1968 Tokachi-oki earthquake[43].

interplate earthquake of >M7.5 in this region is advanced by >4
years. This is because of the faster loading rate indicated by the
observed shortening of the recurrence interval of the repeating
∼M5 earthquakes under the high rate of postseismic slip in
subarea 1 (ref. 41).

The averaged recurrence interval of the earthquakes at the
rupture area of the 1968 Tokachi-oki Earthquake has been
estimated to be ∼97 years based on the past events that occurred
in 1677, 1763, 1856 and 1968 (ref. 42). The maximum slip amount
of the earthquake in 1968 has been estimated to be ∼9 m (ref. 43),
while the plate convergence rate, which is equal to the steady slip
rate at the portion with zero coupling between the subducting and
continental plates, is ∼8 cm per year around the Tohoku
district[44,45]. During the period analysed in this study, the
cumulative postseismic slip on the plate interface reached
∼40 cm near the source region of the 1968 Tokachi-oki
Earthquake within 1 year after the occurrence of the M9

mainshock, which is equivalent to 5 years steady slip at the rate
evident before the Tohoku-oki Earthquake. Therefore, the next
earthquake at the rupture area of the 1968 Tokachi-oki Earthquake
could occur at least 4 years earlier than the ordinary recurrence
interval, if the interplate coupling ratio does not change.

Recurrence intervals of such large interplate earthquakes are
known to have intrinsic irregularities, which could be related to
various factors, for example, stress perturbations due to nearby
seismic and/or aseismic slip events along the plate boundary, to
large intraplate earthquakes, tidal loading effects. Ref. 46 have
recently revealed that the interplate coupling in the northeast
Japan subduction zone shows periodical variations with 2–6 years
period, and large earthquakes may be triggered by the
acceleration of the aseismic slip on the plate interface. It is
possible that the postseismic slip associated with the Tohoku
Earthquake advances the next Tokachi-oki earthquake by more
than 4 years, as the result of the interplay with the newly found
slow slip acceleration cycle. Moreover, the postseismic slip in this
region did not terminate until the end of the analysis period, and
possibly still continues. Thus, the occurrence time advance must
be more than the above estimated value, but further investigation
using the seafloor geodetic observation data[39] will be necessary to
determine the time of advance more accurately.

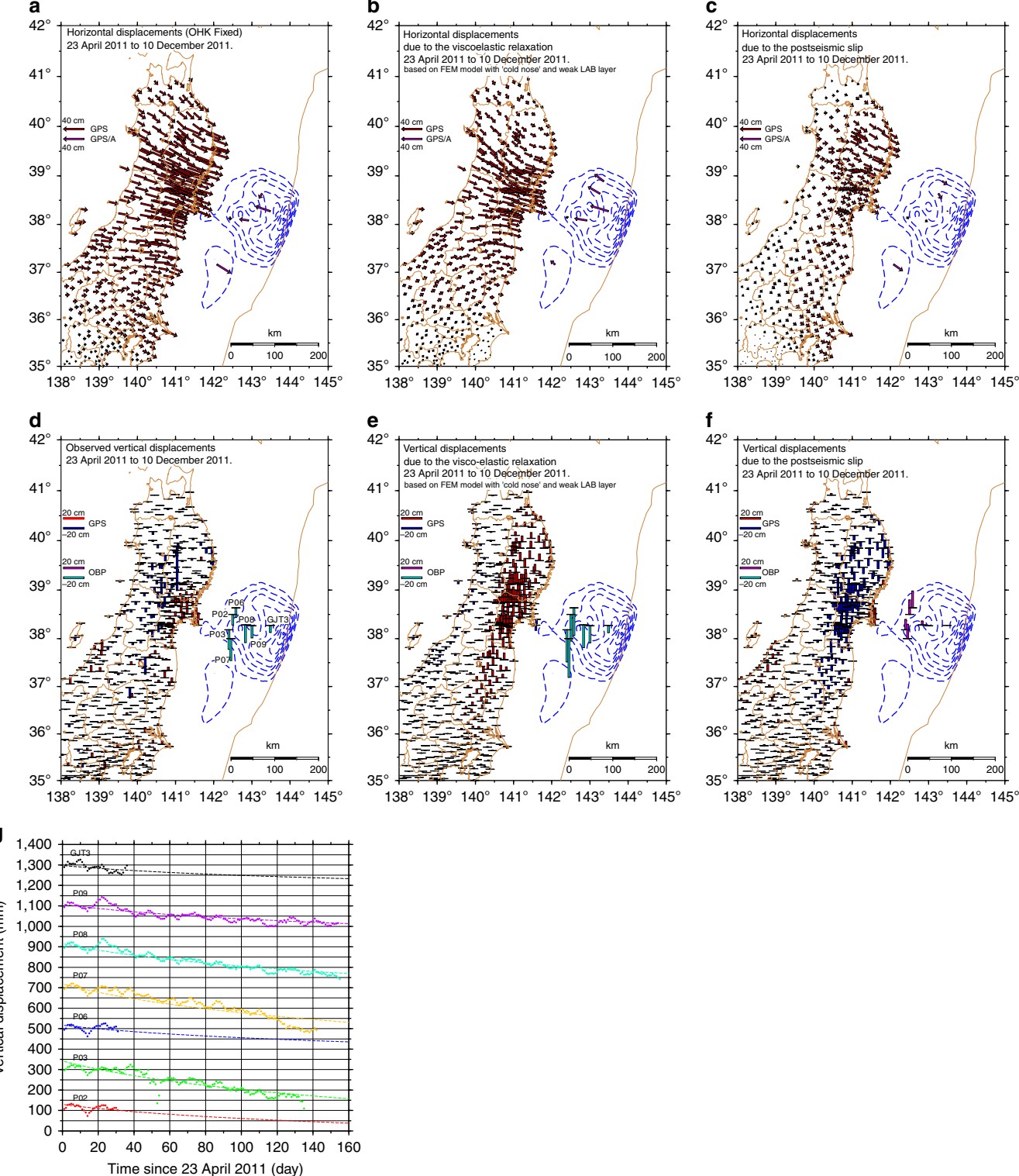

**Figure 6 | Postseismic displacement fields and vertical displacement time series at OBP sites.** (**a**) Horizontal and (**d**) vertical components of observed postseismic displacements from 23 April 2011 to 10 December 2011. Horizontal vectors are in the Okhotsk plate (OHK) fixed reference frame. The blue dashed contours represent the coseismic slip distribution[25]. Site codes of OBP stations (P02, P03, P06, P07, P08, P09 and GJT3) whose time series are exhibited in **g** are shown. (**b**) Horizontal and (**e**) vertical displacements calculated by applying the FEM model[33]. (**c**) Horizontal and (**f**) vertical displacements regarded as being due to the postseismic slip. Note that the scales of the displacements shown in this figure differ among the panels. (**g**) Daily averaged seafloor vertical displacement time series at OBP sites. Dashed lines represent logarithmic curves that are estimated the observed raw data plotted with coloured dots. FEM, finite-element method.

## Methods

**Collection of geodetic data time series.** We used the observations of displacement from the continuous GPS observations from 23 April 2011 to 10 December 2011. The data during the period immediately after the mainshock occurrence until 23 April are excluded in order to minimize the effect of poroelastic relaxation and coseismic deformations associated with several major intraplate (both in the shallow crust and the oceanic slab) earthquakes happened in the period. In the present study, we limited the analysis period to the early stage of the postseismic period, so that we can neglect the effect of the recoupling between the subducting and continental plates, which gives additional complexity in interpreting the deformation data.

GPS data observed at stations managed by the Geospatial Information Authority of Japan (GSI), Tohoku University (TU), Japan Nuclear Energy Safety Organization (JNES) and National Astronomical Observatory (NAO) were analysed to obtain daily station coordinates with uncertainties of ~2 and 5 mm for horizontal and vertical components by applying Precise Point Positioning[47] using GIPSY-OASIS II software. Overall, 383 GPS sites were used in the inversion analysis used to estimate the postseismic slip distribution on the plate interface. It is reasonable to expect an improvement in the spatial resolution of the inversion analysis by including GPS stations of organizations other than the GSI, namely 38 sites are added to 345 GSI's sites, because we can detect the short-wavelength spatial variation in the surface displacement rate field by using the dense GPS observation array. The detailed examination of the improvement of the spatial resolution is presented in ref. 25.

Seven GPSA stations in and around the rupture area of the 2011 Tohoku Earthquake have been in operation under the management of the Japan Coast Guard (JCG) and TU since the occurrence of the earthquake. In the inversion analysis, we used six GPSA stations where the surveys were performed more than twice after the Tohoku Earthquake[33,34].

The 1 Hz sampled OBP records of six sites were used in this study. Raw OBP data are available before 23 September 2011, that is, about beginning two-thirds of the entire analysis period from 23 April 2011 to 10 December 2011. We estimated changes in the seafloor levels using the OBP gauges[48]. As these pressure gauges were installed before the 2011 Tohoku Earthquake, the secular change in the OBP due to sensor drift can be excluded by linear and exponential curve fitting to the records before the mainshock occurrence. Thus, we were able to obtain vertical displacement time series due to the postseismic deformation associated with the 2011 Tohoku Earthquake from these pressure records. These OBP records were shorter than the entire analysis period and therefore, we extrapolated the time series by fitting a logarithmic function to the available postseismic displacement time series (Fig. 6g).

All site coordinates and daily displacement time series are available as Supplementary Data sets 1 and 2.

**Preparation of the inversion analysis..** We estimated the station displacement, which can be regarded as having been caused by the postseismic slip, by subtracting the displacement due to steady plate motion, large aftershocks and viscoelastic relaxation from the observed postseismic displacement time series. Figure 6 shows the displacements due to viscoelastic relaxation and displacements regarded as being due to postseismic slip after the corrections, as well as the observed postseismic displacements.

The effect of the plate motion is corrected by transforming the observed data into the Okhotsk-plate-fixed reference frame by applying the plate motion model of the ITRF2005 (ref. 49). We estimated and removed the coseismic displacements associated with the intraplate earthquakes. The displacements at the observation stations were calculated based on the centroid-moment-tensor (CMT) solutions of the earthquakes during the analysis period, provided by Japan Meteorological Agency.

The viscoelastic deformation was predicted using a novel finite-element method[33]. Their viscoelastic model is based on the realistic structure of Northeast Japan arc-trench system such as the subducting oceanic slab. Burgers rheology was assumed within the viscoelastic layer. The model successfully reproduces not only seafloor but also terrestrial displacements, while no other models succeeded to explain all the onshore and offshore crustal deformation so far. The constructed finite-element method model was tuned such that no landward residual displacement remained when the calculated viscoelastic displacement was subtracted from the observed viscoelastic displacement. This tuning was performed to ensure that no normal faulting motions are required to explain the residual displacements, considered to reflect the postseismic slip on the plate boundary. On the other hand, the postseismic reverse faulting slip would be overestimated when displacements due to viscoelastic relaxation are underestimated[33].

**Estimation of the postseismic slip.** A time-dependent inversion method[6] was used to estimate the postseismic slip distributions associated with the 2011 Tohoku Earthquake from the corrected postseismic crustal deformation data. The postseismic slip distribution was expressed by the dip-slip on the triangulated tessellation of the plate interface geometry[50] in a homogeneous elastic half-space[51]. Strike-slip components were ignored to reduce computation time. The weights of the constraint condition with respect to the spatial and temporal smoothness of the slip distribution, and the Dirichlet-type boundary condition were optimized by minimizing Akaike's Bayesian Information Criterion[52,53].

We undertook a set of computational tests to examine the spatial resolution of the inversion analysis. In these tests, displacements at the geodetic sites were calculated using slip distributions with a checkerboard pattern on the plate interface and were inverted for the slip distribution (Supplementary Fig. 3). The difference between a given slip amount and the estimated one at a triangular element was evaluated. The test was performed using 100 different checkerboard patterns with changing the pattern alignment. The differences between given and estimated slips were averaged to know how well the slip amounts were recovered by the inversion. Here, we regard the slip amounts at the triangular elements with the averaged difference <0.3 m are well constrained. Note that the formal estimation errors (Fig. 1b) exceed 0.3 m in the areas where the averaged slip differences of this test are larger than 0.3 m.

**Selection of the interplate earthquakes.** Interplate seismicity can be independent information regarding the aseismic slip along the plate interface if we assume that the interplate small earthquakes are triggered by the aseismic slip around the fault patches of them. Here we inspect the change in the seismicity rate of the interplate earthquakes identified according to their focal mechanisms. In addition to the thrust faulting earthquakes on the F-net focal mechanisms catalogue[54] and the small repeating earthquakes[29], we included small offshore earthquakes whose mechanism solutions were estimated by ref. 36 based on the seismic waveforms similarities to earthquakes of known focal mechanisms.

**Data availability.** All geodetic data used in this study are included in Supplementary Data sets 1 and 2.

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

## Acknowledgements

We thank J.C.G. for providing the GPSA data. GPS data were provided by the J.N.E.S., G.S.I. and N.A.O. We are grateful to Dr Yusaku Ohta at Tohoku University for raw GPS data processing and providing displacement time series. The present study was supported through the MEXT project, 'Evaluation and disaster prevention research for the coming Tokai, Tonankai and Nankai earthquakes' and by JSPS KAKENHI (15K05260, 20244070, 26000002 and 26109007). We thank three anonymous reviewers for their constructive comments that greatly improved this manuscript. Figures were created using GMT software[55]. We thank Editage (www.editage.jp) for English language editing.

## Author contributions

T.I. performed the inversion analysis and wrote the paper, except those paragraphs that explain the seismological data analysis, which was written by N.U. who also performed the analyses of the repeating earthquake data with W.N. and R.H. performed the OBP observation and processed the raw data to obtain the times series of seafloor level change. M.K. and Y.O. undertook the GPSA measurements and data processing. S.M. collected terrestrial GPS data and processed them to obtain the displacement time series. All authors contributed to the data interpretation.

## Additional information

**Competing financial interests:** The authors declare no competing financial interests.

