## [Peer Review File · Nature Communications]

Reviewer #1 (Remarks to the Author):

This paper addresses the issue of mapping post-seismic deformation that followed the Tohoku-Oki 2011 event.

I have four major problems with this paper:

1- Many references are missing, some of them show that indeed there was two ruptures (Maercklin et al. 2012; Roten et al. 2012; Psimoulis et al., 2014), this is well visible in the seismogram time-series) other proposed a single patch (Koketsu et al. 2011; Suzuki et al. 2011; Yagi & Fukahata 2011; Yue and Lay, 2011).

When the authors compare the co-seismic rupture with post-seismic slip, they cannot ignore the diversity of the co-seismic rupture models. Citing them is a minimum, discussing them and show them on the figures would be good thing to publish a paper in Nat. Geosc.

2- The post-seismic motion is expected to be "around" the coseismic rupture. This is true for all kind of event, it is not usually seen for moderate magnitude event except for Parkfield event (model very well known). See Kim and Dreger, 2008 and Houlie, Dreger and Kim, 2014.

3- I think some of the Figures located in Sup. Mat. should be shown in the main text. Some of them are more interesting than the one in the main text. I let the editor decide on this.

4- What happens between 11.03 and 22.03 (start of the study)? This is during the first days that the postseismic deformation is non-linear and the highest. This must be mentioned in the text somewhere. Even there is no data available.

References

Koketsu, K. et al., 2011. A unified source model for the 2011 Tohoku earthquake, *Earth planet. Sci. Lett.*, 310, 480-487.

Suzuki, W., Aoi, S., Sekiguchi, H. & Kunugi, T., 2011. Rupture process of the 2011 Tohoku-Oki megathrust earthquake (M9.0) inverted from strong-motion data, *Geophys. Res. Lett.*, 38, L00G16, doi:10.1029/2011GL049136.

Yagi, Y. & Fukahata, Y., 2011. Rupture process of the 2011 Tohoku-oki earthquake and absolute elastic strain release, *Geophys. Res. Lett.*, 38(19), L19307, doi:10.1029/2011GL048701.

Yue, H. & Lay, T., 2011. Inversion of high-rate (1sps) GPS data for rupture process of the 11 March 2011 Tohoku earthquake (Mw9.1), *Geophys. Res. Lett.*, 38, L00G09, doi:10.1029/2011GL048700.

Maercklin, N., Festa, G., Colombelli, S. & Zollo, A., 2012. Twin ruptures grew to built up the giant 2011 Tohoku, Japan, earthquake, *Sci. Rep.*, 709(2), 1-7.

Psimoulis, P., Houlie, N., Michel, C., Meindl, M. and Rothacher, M. (2014), Long-period surface motion of the multi-patch Mw9.0 Tohoku-Oki earthquake, *GJI*, 199, 968-980, 10.1093/gji/ggu302

Roten, D., Miyake, H. & Koketsu, K., 2012. A Rayleigh wave backprojection method applied to the 2011 Tohoku earthquake, *Geophys. Res.*

Reviewer #2 (Remarks to the Author):

This paper presents new data and inversion results to constrain the spatial distribution of postseismic slip that occurred following the 2011 Tohoku earthquake. As well as previously-published land-based and ocean-bottom geodetic observations, they use new data from ocean-bottom pressure sensors. The authors conclude that postseismic slip is spatially distinct from the coseismic slip, in contrast to some previously published models. This is an important result for the geophysical community. I found the paper to be generally well written and argued, and recommend publication after the points below are addressed. I consider these to be minor/moderate.

Main points

1. The new data from the OBP gauges should be presented in full and made available for download as a supplementary data set. In particular, the full time series should be shown for all the sites, along with the logarithmic fits that were used to constrain the models.
2. Area C of postseismic slip is poorly resolved and has a high error. Any inferences from this area are therefore poorly constrained. In general, the error plot in Figure 1b should be redrawn with a different colour scale. It is very hard to tell the difference between errors of ~ 0.3 m and ≥ 0.5 m. Perhaps the signal to noise ratio of the slip in patches A, B and C could be quoted? This would give confidence to the result in areas A and B.
3. A more complete exploration of the viscoelastic model space should be presented. Various models are discussed, but the implications on the postseismic slip results should be presented more explicitly. How much difference do different assumptions about rheological structure make to the slip distributions?
4. "Slip partitioning" has a very specific meaning in structural geology, which mean that the title confused me initially. I would suggest not using the phrase "slip partitioning" but instead saying "Spatial separation of coseismic and postseismic slip".
5. I was not convinced by the analysis concerning the 1968 earthquake location. Figure 3 seems to have a high ratio for much of this rupture area suggesting that it did slip postseismically. If it did slip, surely this releases stresses and delays any future earthquake? Furthermore, the variability in inter-event time means that an advance of ~ 5 years would be well within the natural variability from previous cycles.
6. The residuals show significant along-strike horizontal motion. These should be discussed. Perhaps strike-slip motion should be allowed in the inversion for postseismic slip?

Minor points

1. Figure S1 should be in the main manuscript, along with time series from the OBP gauges.
2. Line 78 - state what is "novel" about the FEM model.
3. Line 25 - "eliminate" is perhaps too strong given the likely trade-off between afterslip and viscoelastic deformation.

Reviewer #3 (Remarks to the Author):

The manuscript presents an analysis of postseismic slip on the subduction fault following the 2011 Mw=9 Japan earthquake. The authors use land and offshore geodetic data to constrain the slip distribution, and argue that this postseismic slip pattern is clearly different and complementary to coseismic slip. This result is used to argue for the validity of the "standard" rate-and-state friction model.

The manuscript follows a long line of studies of the 2011 Japan earthquake, and I find it difficult to see the strong originality or novelty of the results presented here, for three main reasons:

(1) The geodetic data analysis used to reveal postseismic slip is only novel in that it combines land and offshore data (but why only 8 month of data when over 4 years are available now ?), corrected for viscoelastic postseismic relaxation. However, several studies, some of them referenced here and others missing, already discuss these points in details. In particular, the authors recognise the non-uniqueness of the viscoelastic postseismic relaxation, but do not include it in their analysis. Why use only one set of predictions (Sun et al., 2014), when several other studies have shown and discussed the variability and uncertainty of this type of model. Cf. in particular the following articles not referenced:

Trubienko et al., *Solid Earth*, 2014

Trubienko et al., *Tectonophysics*, 2013

Hu et al., *Earth, Planets and Space*, 2014

Diao et al., *Geophysical Journal International*, 2014

Yamagiwa, *Geophysical Research Letters*, 2015

All these studies highlight the difficulty and non-uniqueness of determination of postseismic processes (viscoelastic relaxation, afterslip, poroelastic relaxation). Without a more thorough analysis and discussion of the interactions and trade-off between these processes, it seems very hard to define a unique postseismic slip pattern, and thus to be definitive about its characteristics.

(2) The analysis of repeating earthquakes and seismicity rate increase is interesting but not novel. Isn't that what was done by Uchida et al., *GRL*, 2015? How does the manuscript analysis bring new information, other than confirming an increase in seismicity rate around the large coseismic slip patch (which is nothing more than classical coulomb stress increase)?

(3) The argument that postseismic slip surrounds and complement coseismic slip is interesting, but it is presenting as a validation of the standard rate-and-state friction model, without much discussion of the matter at stakes. Are these points (complementarity of slip and rate-and-state model) actually debated? Are there alternate models and data analyses that can be refuted based on the authors work presented here?

Overall, although the study presented in this article is both interesting and serious, I find it difficult to see in what way it differs from yet another analysis of co and postseismic slip and processes following the 2011 earthquake, coming to roughly the same conclusions as most other studies.

Response to the comments of Reviewer #1

Thank you very much for the detailed reviewing and many constructive comments. We revised our manuscript reflecting your and other reviewers' comments and suggestions as below. Your comments are shown in red. The revised manuscript file contains tracked changes, like ~~this~~these except for text formatting and reference numbers.

This paper addresses the issue of mapping post-seismic deformation that followed the Tohoku-Oki 2011 event.

I have four major problems with this paper:

1- Many references are missing, some of them show that indeed there was two ruptures (Maercklin et al. 2012; Roten et al. 2012; Psimoulis et al., 2014), this is well visible in the seismogram time-series) other proposed a single patch (Koketsu et al. 2011; Suzuki et al. 2011; Yagi & Fukahata 2011; Yue and Lay, 2011).

When the authors compare the co-seismic rupture with post-seismic slip, they cannot ignore the diversity of the co-seismic rupture models. Citing them is a minimum, discussing them and show them on the figures would be good thing to publish a paper in Nat. Geosc.

Thank you for your kind suggestion. We cited some of the suggested references and compared their coseismic slip distributions with the postseismic slip distribution in Supplementary Figure 1. Short discussion about the comparison is included in Discussion, lines 300-307.

2- The post-seismic motion is expected to be "around" the coseismic rupture. This is true for all kind of event, it is not usually seen for moderate magnitude event except for Parkfield event (model very well known). See Kim and Dreger, 2008 and Houlie, Dreger and Kim, 2014.

We totally agree. The difficulty to examine the spatial separation between the co- and post- seismic slip for moderate earthquakes comes from the spatial resolution depending on the geometrical coverage of the observation over the source fault. The improvement of the spatial resolution of the inversion analysis (presented in Supplementary Figure 4) points the one of the advantage of this study with using many OBP data.

3- I think some of the Figures located in Sup. Mat. should be shown in the main text. Some of them are more interesting than the one in the main text. I let the editor decide on this.

Because this manuscript was originally prepared to submit to Nature Geoscience that has strict

length limitations (and then transferred to Nature Communications), we could not include those figures in the main text. We have moved Figures S1, S2, and S3 into the main text of the revised manuscript (Figures 1, 3, and 4).

4- What happens between 11.03 and 22.03 (start of the study)? This is during the first days that the postseismic deformation is non-linear and the highest. This must be mentioned in the text somewhere. Even there is no data available.

We added the explanation about the starting time of the analysis in the head of Method as follows: “We obtained the observations of displacement from the continuous GPS observations from 23 April 2011 to 10 December 2011. The data during the period immediately after the mainshock occurrence until 23 April are excluded in order to minimize the effect of poroelastic relaxation and coseismic deformations associated with several major intraplate (both in the shallow crust and the oceanic slab) earthquakes happened in the period. In the present study we limited the analysis period to the early stage of the postseismic period so that we can neglect the effect of the re-coupling between the subducting and continental plates, which gives additional complexity in interpreting the deformation data.”

Followings are the replies and modifications regarding your comments directly written on the manuscript.

Line 14: not only those. Look at Parkfield! Houlie et al., 2014

We inserted a phrase and three references at the first sentence of Introduction. Because the abstract should be shortened to fit the requirement of Nature Communications, we have rewritten the abstract and reflected your comment at Introduction.

Line 24: which model?

Abstract for Nature Communication do not permit to contain references. Therefore, we did not specify the model here, but we described about it in Discussion.

Line 28 and 29: REF

As previously mentioned, we can not contain references in Abstract. These are explicitly referred in Discussion.

Line 31: Why also, you contrast your statement with small events

Examination for the small events goes beyond the aim of this paper. Thus, we have just deleted “also.”

Line 38: we need refs here

References (Refs. 2-7) are inserted here.

Lines 43-44: depth ranges?

References and description about the postseismic slip distribution are added in the first paragraph of Introduction.

Lines 47-54: (Annotations w.r.t references)

References are cited and moved following your suggestions. And the reference numbers are shown one by one.

Lines 55-57: (5 annotations w.r.t wording)

Totally reflected.

Line 56: always! (referring to “terrestrial”)

Continuous GPS observations are performed not only on land, but on sea surface and in the air (sometimes in space, e.g. GRACE).

Line 65: both

“both” is inserted, but “terrestrial” is not erased because of the same reason as the previous one.

Line 70: Did it? Will check you answer this later

We have performed another synthetic test to confirm the advantage of the usage of OBP data for improving the spatial resolution. The result is shown in Supplementary Figure 4 and related discussion are added in the paragraph at lines 320-331.

Lines 73-74: data × 2, REF

All geodetic time series data are contained in the Supplementary Information. References are inserted.

Line 76: it would be better to see this figure in the paper.

Figure S1 was moved into the main document as Figure 1 as suggested.

Line 80: Finite Element Method

Corrected.

Right space of lines 84-90: I am not sure.....

Figures S2 and S3 were moved into the main text as Figures 3 and 4.

Line 96: Really? Data are the same but how about algorithm to model the process?

There are some differences between the processes to estimate the co- and post-seismic slip distributions in fact. For example, we adopted finite triangular subfaults to model the plate interface in this study, while smooth spline functions are applied to express the plate boundary fault by Iinuma et al. (2012). However, such differences are minor compared to the effect of some assumptions such as linear elasticity, infinite half body, the value of Poisson's ratio, and so on. We used the same one for those assumptions.

Lines 105-107: Then why mentioning this?

We would like to state that we cannot distinguish whether no postseismic slip occurs or no seismic patch such that generates the repeating earthquakes exists in the area with no small repeating earthquake. We rewrote these sentences to make them clear.

Lines 115-120: Yes but how about between 11.3 and 23.4? it should be much larger!

We agree with that the postseismic slip before the analysis period should be much larger. However, it is difficult to distinguish that the origins of the crustal deformation as previously explained (see the answer to the fourth comment).

Lines 122-124: you said that at the previous page.

We have removed these sentences.

Lines 127-128: give us some remark here even if you do not show analyses /fig. here

We are sorry that we can not understand what you are mentioning precisely. However, METHOD section is moved into the main text.

Line 131: in your case... which are?

Absence. To clarify it, we added one sentence just after here.

Lines 133-134: obvious

Exactly. However, we would like to remain it for the readers who think it is not obvious.

Figure 3: iso depth lines!

We drew the iso-depth contour of the plate boundary in new Figure 6 (Number of the figure is changed due to the movement of the figures from the supplementary materials into the main text).

Thank you again for your detailed reviewing.

Sincerely,

Takeshi Iinuma

Response to the comments of Reviewer #2

Thank you very much for the detailed review and many constructive comments. We revised our manuscript reflecting your and other reviewers' comments and suggestions as below. Your comments are shown in red. The revised manuscript file contains tracked changes, like ~~this~~these except for text formatting and reference numbers.

This paper presents new data and inversion results to constrain the spatial distribution of postseismic slip that occurred following the 2011 Tohoku earthquake. As well as previously-published land-based and ocean-bottom geodetic observations, they use new data from ocean-bottom pressure sensors. The authors conclude that postseismic slip is spatially distinct from the coseismic slip, in contrast to some previously published models. This is an important result for the geophysical community. I found the paper to be generally well written and argued, and recommend publication after the points below are addressed. I consider these to be minor/moderate.

Main points

1. The new data from the OBP gauges should be presented in full and made available for download as a supplementary data set. In particular, the full time series should be shown for all the sites, along with the logarithmic fits that were used to constrain the models.

Thank you for your suggestion. We have attached the full time series data in Supplementary Information that are used in this study.

2. Area C of postseismic slip is poorly resolved and has a high error. Any inferences from this area are therefore poorly constrained. In general, the error plot in Figure 1b should be redrawn with a different colour scale. It is very hard to tell the difference between errors of ~ 0.3 m and ≥ 0.5 m. Perhaps the signal to noise ratio of the slip in patches A, B and C could be quoted? This would give confidence to the result in areas A and B.

Colour scale for estimation error is reconstructed as shown in Figure 2.

3. A more complete exploration of the viscoelastic model space should be presented. Various models are discussed, but the implications on the postseismic slip results should be presented more explicitly. How much difference do different assumptions about rheological structure make to the slip distributions?

We have added a paragraph about this topic in Discussion as following:

The rheological structure and parameters estimated by Ref. 33 may not be uniquely determined, but are well constrained. The viscosities controlling both the spatial distribution and temporal variation of the deformations were constrained by the time constants of decay of the observed displacement time series. The elastic thicknesses of the oceanic slab and the continental lithosphere are sensitive to the horizontal displacement pattern. Thicker oceanic slab reduces the, landward motion on the seafloor, whereas increases the seaward displacement on land. The increase of the thickness of continental lithosphere enlarges the seaward motion on land. Too large viscoelastic displacement results in the over corrections of the observed postseismic displacement field and the large normal fault type postseismic slip on the fault is required but the substantial normal faulting postseismic slip is physically unrealistic. Such trial-and-error to constrain the viscoelastic deformation model based on terrestrial and seafloor geodetic observation data were thoroughly performed by Refs. 33 and 46, therefore we concluded that their model is the most reliable model for the present.

4. "Slip partitioning" has a very specific meaning in structural geology, which mean that the confused me initially. I would suggest not using the phrase "slip partitioning" but instead saying "Spatial separation of coseismic and postseismic slip".

Thank you very much for your kind suggestion. We have modified "partitioning" into "separation" at the title and related portion in the manuscript.

5. I was not convinced by the analysis concerning the 1968 earthquake location. Figure 3 seems to have a high ratio for much of this rupture area suggesting that it did slip postseismically. If it did slip, surely this releases stresses and delays any future earthquake? Furthermore, the variability in inter-event time means that an advance of ~5 years would be well within the natural variability from previous cycles.

High ratio of the thrust earthquake activity mainly depends on moderate ~ small earthquakes. It means that the postseismic slip inside the rupture area of the 1968 earthquake is not necessarily large, that is, we can not clearly document that the slip inside the rupture area delays the future earthquake.

With respect to the last half of above comment, we inserted the following discussion at lines 358-370 in Discussion.

Recurrence intervals of such large interplate earthquakes are known to have intrinsic irregularities, which could be related to various factors, e.g., stress perturbations due to nearby seismic and/or aseismic slip events along the plate boundary, to large intraplate earthquakes, tidal loading effects. Ref. 54 have recently revealed that the interplate coupling in the northeast Japan subduction zone shows periodical variations with 2 ~ 6 years' period, and large earthquakes may be triggered by the acceleration of the aseismic slip on the plate interface. It is possible that the postseismic slip associated with the Tohoku Earthquake advances the next Tokachi-oki earthquake by more than 4 years, as the result of the interplay with the newly found slow slip acceleration cycle. Moreover, the postseismic slip in this region did not terminate until the end of the analysis period, and possibly still continues. Thus, the occurrence time advance must be more than the above estimated value, but further investigation using seafloor geodetic observation data ⁴⁷ will be necessary to determine the time of advance more accurately.

6. The residuals show significant along-strike horizontal motion. These should be discussed. Perhaps strike-slip motion should be allowed in the inversion for postseismic slip?

The trench parallel residuals may reflect several causes. Short explanation is added in "Postseismic slip based on geodetic data" subsection in Result.

Minor points

1. Figure S1 should be in the main manuscript, along with time series from the OBP gauges.

Figure S1 is moved into the main text and time series of OBP gauges are shown.

2. Line 78 - state what is "novel" about the FEM model.

Their viscoelastic model is based on the realistic structure of Northeast Japan arc-trench system, and reproduces not only seafloor but also terrestrial displacements well, while no other model succeeded to explain crustal deformation with using realistic structure in the present as written in the body text.

3. Line 25 - "eliminate" is perhaps too strong given the likely trade-off between afterslip and viscoelastic deformation.

We agree with your comment, then “subtract” is used instead of eliminate, here.

Thank you again for your detailed review.

Sincerely,

Takeshi Inuma

Response to the comments of Reviewer #3

Thank you for the reviewing and giving comments. We revised our manuscript reflecting your and other reviewers' comments and suggestions as below. Your comments are shown in red. The revised manuscript file contains tracked changes, like ~~this~~these except for text formatting and reference numbers.

The manuscript follows a long line of studies of the 2011 Japan earthquake, and I find it difficult to see the strong originality or novelty of the results presented here, for three main reasons:

(1) The geodetic data analysis used to reveal postseismic slip is only novel in that it combines land and offshore data (but why only 8 month of data when over 4 years are available now ?), corrected for viscoelastic postseismic relaxation. However, several studies, some of them referenced here and others missing, already discuss these points in details. In particular, the authors recognise the non-uniqueness of the viscoelastic postseismic relaxation, but do not include it in their analysis. Why use only one set of predictions (Sun et al., 2014), when several other studies have shown and discussed the variability and uncertainty of this type of model. Cf. in particular the following articles not referenced:

Trubienko et al., Solid Earth, 2014

Trubienko et al., Tectonophysics, 2013

Hu et al., Earth, Planets and Space, 2014

Diao et al., Geophysical Journal International, 2014

Yamagiwa, Geophysical Research Letters, 2015

All these studies highlight the difficulty and non-uniqueness of determination of postseismic processes (viscoelastic relaxation, afterslip, poroelastic relaxation). Without a more thorough analysis and discussion of the interactions and trade-off between these processes, it seems very hard to define a unique postseismic slip pattern, and thus to be definitive about its characteristics.

We agree with that the viscoelastic component of the postseismic deformation can not be determined uniquely. However, the model of Sun et al. (2014) is the only and best one to account for the terrestrial and seafloor geodetic data at present.

For instance, Trubienko et al. (2013 and 2014) do not use the seafloor displacements to construct their rheological structure models, and the landward seafloor movements have never occurred in the deformation pattern along the trench normal direction calculated by them. Diao et al. (2014) also have not applied postseismic seafloor displacement, even though they applied the seafloor data to estimate the coseismic slip distribution.

It seems like that Yamagiwa et al. (2015) succeeded to construct the viscoelastic structure model

and estimated the postseismic slip distribution with applying available terrestrial and seafloor geodetic data, but they adopted simple layered structure for the rheological model. Such layered structure is too simple to express the real structure at the subduction zones. It should be noted that we also have already succeeded to account for the landward displacements on the seafloor by applying a layered structure in 2012 (Iinuma et al. (2012, JpGU Meeting; 2012, AGU Fall Meeting; 2013, IAG Scientific Assembly), because it is much easier to construct the rheological model by applying simple layered structure than to do by applying realistic structure such that includes subducting slab and the contrast between the viscosities in the wedge mantle and oceanic asthenosphere under the constraint that no westward displacements are remained when we subtracted the viscoelastic displacements from the observed ones.

Hu et al. (2014) evaluated the effect of poroelastic rebound, but their model is time-independent. Furthermore, the landward displacements on the seafloor reproduced by Hu et al. (2014) are not enough large to remain no westward residual.

In conclusion, there is no appropriate model to eliminate the viscoelastic relaxation effect from the observed postseismic deformation data other than the model of Sun et al. (2014). Short discussion about this matter is inserted in the body text.

(2) The analysis of repeating earthquakes and seismicity rate increase is interesting but not novel Isn't that what was done by Uchida et al., GRL, 2015? How does the manuscript analysis bring new information, other than confirming an increase in seismicity rate around the large coseismic slip patch (which is nothing more than classical coulomb stress increase)?

There is no paper "Uchida et al., GRL, 2015." Is it Uchida and Matsuzawa (2013, EPSL)? If so, new information brought by this study is the quantitative comparison between the postseismic slip estimated from the small repeating earthquake analysis and from the geodetic data inversion with unifying the analysis periods of seismic and geodetic data.

(3) The argument that postseismic slip surrounds and complement coseismic slip is interesting, but it is presenting as a validation of the standard rate-and-state friction model, without much discussion of the matter at stakes. Are these points (complementarity of slip and rate-and-state model) actually debated? Are there alternate models and data analyses that can be refuted based on the authors work presented here?

We think that the validation of the standard rate- and state-dependent friction law is still a topic of debate in nature, because the spatial resolutions of the estimation of co-, post-, and/or inter-seismic slip distributions based on seismic and geodetic data are not perfect. Especially, with respect to the

mega-thrust earthquakes that occur beneath the seafloor, we (all researchers) should not conclude prematurely. In this context, the argument of Johnson et al. (2012) should not be overlooked as many studies have cited their work (e.g., Shirzaei et al., 2014, EPSL; Tassara et al., 2016, Tectonophysics, Johnson et al., 2016, GRL; Hu et al., 2016, JGR; Lozos et al., 2015, GRL; Jolivet et al., 2015, GRL; Graham et al., 2014, GJI). Therefore, we added several sentences introducing the debate on this topic in Introduction.

Thank you again for your review.

Sincerely,

Takeshi Inuma

Reviewer #1 (Remarks to the Author):

A. The authors discuss the potential use of geodetic seafloor data for a period following the Tohoku event.

B. This work is original but lack a bit of perspective on previous work and is still not really showing how important these data are to improve our resolution

C. Data are of good quality but there is few description of the uncertainties in the text.

D. See above

E. Not sure. It would have been nice to test different mesh for the FEM. The signal is however strong and I suspect it would not change much the results.

F. see annotated PDF. The introduction is somewhat not good. Authors throw at us sentence sometimes not connected to the text, etc. easy to improve but it should be done.

G. Nothing to say about this.

H. See point F.

Reviewer #2 (Remarks to the Author):

I have reviewed the revised manuscript, paying particular attention to the changes that I recommended as a reviewer.

I thank the authors for their thoughtful responses, which address all my concerns. The publication of the time series of data is particularly valuable for the community.

I am therefore happy that the manuscript should be published with no further substantial changes.

Reviewer #3 (Remarks to the Author):

Overall, the authors responses and modifications address my comments in a satisfactory manner.

I am not 100% convinced that the manuscript presents major novelty in its analysis as requested by Nature editorial line, but this is al decision that I leave to the editors on basis of the other reviews.

Response to the comments of Reviewer #1

Thank you very much for the detailed reviewing again. We revised our manuscript reflecting your comments and suggestions as below. Your comments are shown in red. The revised manuscript file contains tracked changes, like this except for text formatting and changes in the reference numbers.

Lines 17-18: or to the stress properties? mantle viscosity?

The most important factor other than the frictional property that controls slip behavior on the fault must be material property surrounding the faults including the viscous response, which causes heterogeneous stress distribution. Thus, it is inserted.

Line 34: rewrite this. it does sound funny

“but” is certainly inappropriate as you pointed. We have replaced it with “and.”

Line 40: explain origin of that stress

The tectonic loading is the primary in this context, even though seismic and aseismic events and viscoelastic and poroelastic responses due to such events can produce the stress change. There is no end to list all candidates, therefore, we would like to display only the primary one here.

Line 48: give us the dates of the latest here?

A description about the most recent M7.5 class event is inserted.

Line 49: This is too vague. why are you citing this?

We have not commented on specific feature from subset of previous study. We rather intended to the general feature of the earthquake from many studies. To clarify this we rewrote the sentence.

Lines 51-54: How many events are we talking about? 1, 2, 10? there are large discrepancies between models, you should mention this.

We are talking about the Tohoku-oki earthquake. There is surely large variation in the models, but the feature that we mentioned here is common one. A phrase to mention the differences are added.

Lines 57-59: Why is that important for your work?

Because the breakdown of the spatial separation of the slip behavior means the doubt regarding the validity of application of the empirical rate- and state-dependent friction law to the faults in

nature, as added in the body text.

Line 68: as always....

GPS observations are performed not only on land but also on the sea surface, in the air, and in the space. For example, GOCE and GRACE mount GPS receivers. The seafloor geodetic observations mentioned here also use GPS in the measurement above the sea. To contrast with the seafloor GPS data, we retained the word “terrestrial” here.

Line 74: remove this word. We assume you do so.

We removed the word “appropriately” as suggested.

Line 78: The source region is really diverse in that case. You should draw conclusions about this. That may be the most interesting result.

We added short description about the diversity of the frictional property and slip behavior in the source region.

Line 82: from whom?

The organizations that performed GPS observations are described in the next paragraph. We also rephrased this sentence.

Line 94: Tell us about uncertainties here?

We inserted a phrase mentioning it as “with uncertainties of about 2 mm and 5 mm for horizontal and vertical components.”

Lines 95-99: rewrite this. Expand it, etc. it is not clear. you expect to increase troposphere estimation? please explain.

Line 98: Why?

We clarified why we expect the improvement of resolution compared to the analysis based only on GSI's data.

Line 105: in that case and in general: be more specific

Brief explanation about the pressure sensors is inserted the top of this paragraph. You can find detailed examination about it in Ref. 36.

Line 119: ??? because you did not use any coseismic data? it is not really clear here.

We are sorry that paragraphs around here include some redundant description. We rewrote them.

Line 131: remove

We have removed it as suggested.

Line 243: in 2015 neither. just joking. is this really a surprise as you do not consider the data for the first 12 days that followed the event?

We excluded the data for the first 42 days, because the displacement data immediately after the main shock contains viscoelastic relaxation, postseismic slip, poroelastic rebound, and aftershocks in the hanging and foot walls of the plate interface fault, and because the seafloor coseismic displacement based on the GPS/A measurement includes all of them with some pre-seismic deformation. It is so difficult to distinguish the contributions of these factors that we limited the analysis period after 23 April, 2011. Further study is necessary to estimate the postseismic slip distribution immediately after the earthquake.

On the other hand, postseismic slip on the plate boundaries in subduction zones often continue several months to years. Therefore, the spatial separation between the co- and post-seismic slip during the period in this study is important, even though we excluded the immediately after the main shock.

Lines 272-281: I think most the discussion could find its right place in the results section.

We have moved several paragraphs from Discussion section to Results section.

REF.5: Houlié

I'm very sorry for the misspelling.

Fig.1 (a): where is OHK?.

“OHK” stands for Okhotsk Plate. We put it in parentheses in the caption of Figure 1.

Thank you again for your detailed reviewing.

Sincerely,

Takeshi Iinuma